# A Comparison of Vaping Behavior, Perceptions, and Dependence among Individuals Who Vape Nicotine, Cannabis, or Both

**DOI:** 10.3390/ijerph191610392

**Published:** 2022-08-20

**Authors:** Savreen K. Saran, Kalin Z. Salinas, Jonathan Foulds, Övgϋ Kaynak, Brianna Hoglen, Kenneth R. Houser, Nicolle M. Krebs, Jessica M. Yingst, Sophia I. Allen, Candace R. Bordner, Andrea L. Hobkirk

**Affiliations:** 1Department of Psychiatry and Behavioral Health, Penn State College of Medicine, 500 University Drive, Hershey, PA 17033, USA; 2Department of Public Health Sciences & Penn State Center for Research on Tobacco and Health, Penn State College of Medicine, 500 University Drive, Hershey, PA 17033, USA; 3School of Behavioral Sciences & Education, Penn State Harrisburg, 777 West Harrisburg Pike, Middletown, PA 17057, USA

**Keywords:** nicotine, cannabis, vaping, dependence, electronic cigarette, marijuana, tobacco

## Abstract

Background: Electronic delivery systems (e.g., vapes, e-cigarettes) are now popular modes of cannabis and nicotine administration that are often used by the same individuals; however, we still know little about dual nicotine and cannabis vaping. Materials & Methods: An online convenience sample of adult nicotine and/or cannabis vape users residing in the United States completed a 60 min survey on sociodemographic characteristics, cannabis and/or nicotine vape use behaviors and dependence, reasons for vape use, and perceptions of benefits and harms. After data cleaning, we compared dual vs. nicotine-only and cannabis-only vape users with univariate statistics and step-wise hierarchical linear regression analyses. Additionally, we assessed the factor structure, internal consistency, and criterion and convergent validity of the Penn State Cannabis Vaping Dependence Index (PSCVDI). Results: The final sample included 357 dual, 40 cannabis, and 106 nicotine vape users. Compared to nicotine- and cannabis-only vapers, dual vapers started using their nicotine and cannabis vapes at a younger age (*p* < 0.001), used them for more years (*p* < 0.001), and were less likely to use their nicotine vape to replace combustible cigarettes (*p* = 0.047). Dual users vs. single-substance users did not have significantly higher nicotine or cannabis vape dependence scores after controlling for sociodemographic and use behaviors. The PSCVDI showed adequate validity for measuring cannabis vape dependence. Conclusions: This survey is the first to highlight important differences in vape use behaviors and reasons for use between dual vs. cannabis- and nicotine-only vape users.

## 1. Introduction

Shifts in cannabis policy, including the decriminalization, medicalization, and legalization of cannabis, have contributed to an increased prevalence in the use and rates of cannabis use disorders across the United States (U.S.) [1,2]. In 2020, 49.6 million people in the U.S. used cannabis in the past year, making it the most commonly used illicit substance in the country [3]. The dual use of cannabis and nicotine is also common [4,5]. The 2020 National Survey on Drug Use and Health (NSDUH) shows that, among U.S. residents 12 or older, 35.7% of past 30-day cigarette smokers reported the use of cannabis in the past 30 days, and 33.7% of past 30-day cannabis users reported the use of cigarettes in the past 30 days [3]. The dual use of combustible cannabis and nicotine is associated with greater nicotine and cannabis dependence, worsened treatment outcomes, and a greater risk of respiratory problems from toxicant exposure [4,6,7,8,9].

Globally, electronic delivery systems (e.g., vapes, vape pens, and e-cigarettes) that aerosolize oils or liquids have become a popular mode of cannabis and nicotine administration [10,11]. In the U.S., 5.1% of adolescents aged 12 to 17, 11.7% of young adults aged 18 to 25, and 2.4% of adults aged 26 and older used nicotine e-cigarettes in the past 30 days according to the 2020 NSDUH [3]. Nicotine vapes are typically used with e-liquids containing solvents (e.g., propylene glycol and vegetable glycerin), nicotine base or salt formulations, and flavor additives [12]. Cannabis vapes primarily contain cannabis-derived ingredients, including tetrahydrocannabinol (THC), cannabidiol (CBD), and terpenes, along with many other constituents used as fillers [13]. The Monitoring the Future Survey found that, in 2021, 18% of 12th graders and 12% of 10th graders reported vaping cannabis in the past year [14]. Among adults in the Behavioral Risk Factor Surveillance System, 2% reported past 30-day cannabis vaping [15]. Cannabis and nicotine vaping has been associated with a higher frequency of polysubstance use, including combustible cigarettes, alcohol, and illicit or prescription drug misuse [15,16,17].

Nicotine and cannabis vapes can be used concurrently (i.e., in the same defined time period), used sequentially (i.e., one after the other), or co-administered (i.e., in the same e-liquid) [5]. However, there are limited population-level data on the prevalence of nicotine and cannabis vape dual use. Youth who use nicotine vapes have almost five-times-higher odds of also using a vape with cannabis [18], and adults reporting the use of cannabis vapes are more likely to also use nicotine vapes [15]. In a survey of 112 adult dual users, participants reported that they use both nicotine and cannabis vapes in specific places (e.g., home, work, parties, etc.) with specific people (e.g., friends, family, etc.) and often reported that one substance enhanced the effects of the other [19]. For these dual users, the ease, convenience, and pleasant subjective effects of the vape experience were the most common reasons reported for using, while harm reduction was among the least common reasons [19].

As cannabis becomes more popular and less restricted in the U.S., there are growing concerns about dependence, especially for cannabis vapes with flavor additives and discrete designs that can promote more frequent use compared to combustible forms [20]. Yet, the risk for cannabis vape dependence among cannabis-only and dual users is still unknown. The dependence on one substance may confer risk for dependence on another substance through similar neural and behavioral pathways or shared risk factors, such as enhanced dopamine transmission, more frequent use, or similar individual traits or socioenvironmental circumstances [21]. One challenge to assessing cannabis vape dependence is the lack of validated measures. Several assessment tools have been validated to measure nicotine e-cigarette dependence, including the Penn State Electronic Cigarette Dependence Index (PSECDI) and the E-cigarette Dependence Scale (EDS) [22,23]. There are several validated tools for measuring marijuana dependence that were designed primarily for combustible forms of cannabis, including the Severity of Dependence Scale [24] and the criteria outlined in the Diagnostic and Statistical Manual of Mental Disorders Fifth Edition (DSM-V) [25]. Identifying valid measures for assessing dependence on cannabis vapes is critical to assessing the public health impact of cannabis vaping and developing effective treatments.

Given the potential additive risks of nicotine and cannabis vaping and the relatively limited knowledge on this emerging substance use trend, the current study aimed to characterize the sociodemographic characteristics and vape use behaviors, motivations, perceptions, and dependence levels among adults using both nicotine and cannabis vapes in comparison to those who vape only one substance. To our knowledge, this is the first comprehensive survey comparing dual-drug vape use to single-drug vape use. This investigation adds to existing studies that compared nicotine and cannabis vape use among dual users and assessed differences between vape use and combustible product use [5,19]. To complete our aim of measuring cannabis vape dependence, we also assessed the psychometric properties of an adapted version of the PSECDI, called the Penn State Cannabis Vaping Dependence Index (PSCVDI), and an adapted version of the DSM-V Cannabis Use Disorder criteria among the current sample.

## 2. Materials and Methods

This was a cross-sectional online survey of adults 21 years or older in the U.S. who reported using e-cigarettes or vapes containing nicotine or cannabis in the past 30 days. We recruited participants using Amazon’s Mechanical Turk (MTurk). MTurk is an online labor market that uses a crowdsourcing platform to recruit diverse samples of anonymous participants in the U.S. for online tasks. Data collected via MTurk have been shown to be valid and reliable for social science research [26]. M-Turk has already shown to be an efficient and convenient method for recruiting e-cigarette users for survey research [27], in part because this labor market includes more substance users than the general population [28,29,30]. The main disadvantage of using such online samples is the possibility of inattentive, fake or automated responses [29]. Thorough efforts to screen out such respondents are required, and these are described below.

### 2.1. Procedures

The online survey was administered in two waves with unique participants in each wave. The first wave was posted in June through September of 2020, and the second wave was posted in August through September of 2021. Slight modifications were made to the second wave of the survey to shorten the survey and improve question clarity. The survey was visible to all MTurk workers with the title, “Measuring Attitudes and Use Behaviors of Tobacco Products and Electronic Cigarettes,” along with the description, “Do you use electronic cigarettes, vapes, or tobacco? Researchers at Penn State Hershey are conducting a survey to learn more about user’s behaviors and attitudes towards these types of products.” Only workers with job approval ratings of 98% or greater were allowed to complete the survey. The approval rating is assigned by MTurk based on how many jobs the worker successfully completed and provides a metric of their data and performance quality. Interested workers clicked on a link that directed them to a survey on the Research Electronic Data Capture (REDCap) secure web-based platform used for data collection and management and hosted by the Penn State Health Milton S. Hershey Medical Center and the Penn State College of Medicine [31]. Participants provided implied informed consent by continuing with the survey after being presented with a summary explanation of the research. In choosing to continue with the survey, the participants also certified that they had used an e-cig or vape in the past 30 days, were 18 years of age or older, could read and write in English, and were not a member of the European Union General Data Protection Regulation. Workers were compensated USD 3 in wave 1 and USD 5 in wave 2 for completing the survey. The compensation was adjusted for wave 2 after learning that the survey took longer than expected to complete (30–60 min). All procedures were approved by the Penn State College of Medicine Institutional Review Board (STUDY# 14815).

### 2.2. Participants

Self-identified past 30-day nicotine e-cigarette or cannabis vape users were eligible for the study. Eligibility for the survey and assignment as a nicotine vaper, cannabis vaper, or dual vaper were determined with the following questions in wave 1: “How often have you used a cannabis, CBD, or THC vape in the last 30 days?” and “How often have you used a nicotine electronic cigarette product in the last 30 days” with responses “Every day, most days, some days, not at all”. We changed the wording of the questions slightly to improve clarity in wave 2: “Have you used a vape pen or e-cigarette that contains nicotine in the past 30 days?” and “Have you used a vape pen or e-cigarette that contains cannabis, marijuana, THC, or CBD in the past 30 days?” with the response options “Yes or No.” Participants who endorsed the use of either a nicotine vape or a cannabis vape in the past 30 days were categorized as a single user, and those who endorsed using both in the past 30 days were categorized as a dual user. Age was assessed with the question, “How old are you?”, and participants at least 18 years of age were eligible for the survey, although those under 21 years were later excluded (see Data Analysis).

### 2.3. Measures

The survey battery included measures developed by our team and adapted from standard tobacco assessment protocols to assess nicotine and cannabis vape use. The full survey battery for wave 2 is provided in Appendix A. The surveys used for the current analysis included vape use characteristics, perceived vape benefits and harms, vape use motivations, and vape dependence. The vape surveys were formatted into two sets to inquire specifically about nicotine or cannabis vape use, and dual users completed both sets of surveys to allow for comparisons across vape types. To assist with the psychometric validation of the cannabis vape dependence measure, we also included assessments on general cannabis dependence (not cannabis vapes specifically).

#### 2.3.1. Sociodemographic Characteristics

The sociodemographic questionnaire included questions about race, ethnicity, education, employment, relationship status, sexual orientation, and household income.

#### 2.3.2. Cigarette and Vape Use History and Characteristics

Cigarette and vape use history and product characteristics were assessed using measures developed by our team and adapted from the Population Assessment of Tobacco and Health Study [32], the PhenX Toolkit tobacco and e-cigarette protocols [33], and the Behavioral Risk Factor Surveillance System (BRFSS) [34]. The survey inquired about age at initiation and regular use, poly-tobacco use, tobacco product type, place of purchase, flavor preferences, and current or prior cessation attempts. Questions developed by our team inquired about perceived product safety, weight management motivations, and stealth vaping (i.e., vaping discreetly in places where it is restricted or socially unacceptable).

#### 2.3.3. Smoking Status

Cigarette smoking status was determined using the BRFSS items, “Have you smoked at least 100 cigarettes in your entire life?”, “Do you now smoke every day, some days, or not at all?”, and “How long has it been since you last smoked cigarettes regularly?” [34].

#### 2.3.4. Reasons for Vape Use and Perceived Benefits and Harms

The survey included 11 reasons for using electronic cigarettes or vapes and 30 perceived benefits and 20 perceived harms of using these products. Participants were asked to check all that applied to them and were given an open-ended response option to provide reasons for use. The items for these questions were compiled by our team based on the PATH Study and a literature review of existing studies that inquired about use motivations and perceptions [35,36]. The potential reasons for use and perceived benefits spanned health and social benefits compared to combustible cigarettes, taste and sensory experience, and social and psychological effects. Potential concerns included a lack of effectiveness for smoking cessation, health and addiction concerns, unpleasant sensory experiences, and uncertainty about ingredients.

#### 2.3.5. Cannabis Use

All cannabis use (not just vaped) was assessed with the Daily Sessions, Frequency, Age of Onset, and Quantity of Cannabis Use Inventory (DFAQ-CU) [37]. The DFAQ-CU had good reliability and convergent, predictive, and discriminant validity among over 2000 cannabis users in the United States [37]. During the measure’s development, the DFAQ-CU demonstrated a 6-factor structure that included daily sessions, frequency, age of onset, marijuana quantity, cannabis concentrate quantity, and edibles quantity [37].

#### 2.3.6. Nicotine Vape Dependence

Nicotine vape dependence was measured using the Penn State Electronic Cigarette Dependence Index (PSECDI). PSECDI has normative data on over 3600 e-cigarette users and has shown construct validity in that the scores are related to the nicotine concentration of liquids used [22]. The total scores on the PSECDI range from 0 to 20, where higher scores relate to higher dependence severity [22]. The scale includes two items about sleep that we summed into a single item for the reliability and factor analysis, making it a 9-item scale: “Do you sometimes awaken at night to use your electronic cigarette?” and “If yes, how many nights per week do you typically awaken to use your electronic cigarette?”. Levels of dependence were categorized, according to the PSECDI measure validation, as not dependent (0–3), low dependence (4–8), medium dependence (9–12), and high dependence (≥13) [22]. The index comprises two factors, with the first two questions (use times per day and use after waking) comprising one factor consistent with the Heaviness of Smoking Index used to assess cigarette dependence [38]. PSECDI had good convergent validity with the E-cigarette Dependence Scale among 1009 e-cigarette users (r = 0.78) and among exclusive e-cigarette users (r = 0.71) [23]. Lower PSECDI total scores predicted a higher likelihood of quitting e-cigarette use two and a half years later [39]. The one-month test-retest reliability coefficient among 5–6-month 36 mg/mL nicotine concentration e-cigarette users in an RCT was +0.85 [40].

#### 2.3.7. Cannabis Vape Dependence

For cannabis vape use, we adapted the PSECDI by changing “electronic cigarette” in each item to “cannabis vape,” and we refer to this version as the Penn State Cannabis Vaping Dependence Index (PSCVDI). We used the same scoring as the PSECDI and assessed the factor structure, reliability, and validity for cannabis vaping, as described in the Data Analysis section. 

#### 2.3.8. Cannabis Vape Use Disorder

The DSM-V includes Cannabis Use Disorder, which replaces the “dependence” disorder of the Fourth Edition [25]. This disorder is diagnosed if at least 2 of 11 criteria are met in the same 12-month period. These criteria include: (1) Using cannabis in larger amounts or over a longer period than was intended; (2) Persistent desire or unsuccessful efforts to cut down or control cannabis use; (3) A great deal of time is spent in activities necessary to obtain, use, or recover from cannabis; (4) Craving or a strong desire to use cannabis; (5) Recurrent cannabis use resulting in a failure to fulfill major role obligations at work, school, or home; (6) Continued cannabis use despite persistent or recurrent social or interpersonal problems caused by use; (7) Giving up or reducing important social, occupational, or recreational activities because of cannabis use; (8) Continued use in situations where it could be physically hazardous; (9) Continued use despite physical or psychological problems caused or exacerbated by cannabis use; (10) Tolerance defined by need for increased amounts to achieve the same effects or diminished effects with the same amount; (11) Presence of withdrawal symptoms. For the current survey, we assessed each of these 11 domains with wording that asked specifically about cannabis vaping with the response options “Yes” or “No”. Each item endorsed (one point each) was summed to create a total score ranging from 0 to 11. We assessed the presence of withdrawal symptoms and using a cannabis vape to reduce withdrawal symptoms separately. Endorsing either of these items contributed one point to the total DSM score. We also included follow-up questions designed by our research group inquiring about the symptom severity for each item endorsed. These were not included in the total score.

#### 2.3.9. Cannabis Dependence

Cannabis vapers also completed the 5-item Severity of Dependence Scale (SDS) to measure the degree of psychological dependence on cannabis in general, not vapes specifically [24]. Response options range from 0 (never/almost never) to 3 (always/nearly always) and, for item 5 only, from 0 (not difficult) to 3 (impossible). The responses were summed to create a total score ranging from 0 to 15, with higher scores indicating more severe cannabis dependence. Psychometric properties of the SDS were assessed among 577 Dutch frequent adult cannabis users and showed a single factor with good internal consistency (α = 0.70) [24]. The criterion validity with the DSM-IV-determined diagnosis for cannabis dependence was low, however, with low sensitivity (61.3%) and specificity (63.5%) for differentiating dependence at the optimal cut-off of SDS ≥ 4.

The survey also included the single item, “How addicted are you to cannabis vaping, with response options ranging from 0 (Not at all addicted) to 100 (Extremely addicted).

#### 2.3.10. Validity Check Items

Several bogus questions were included to assess for attentiveness and valid responding, including “I have been to every country in the world”, “I have never brushed my teeth”, and “I am paid biweekly by leprechauns” [41]. The following human intelligence question was also included: “Type *I’m paying attention* in the text box” [42]. Self-reported effort and engagement were assessed by the items, “I have carefully read every survey item“ and “I could have paid closer attention to the items than I did,” to which respondents could agree or disagree [41].

### 2.4. Data Cleaning

We received 649 unique survey respondents in wave 1 and 381 in wave 2. The primary disadvantage of data collection using online labor markets such as MTurk is the potential for workers to provide incomplete data or inattentive or random responses, along with the use of automated algorithms to complete surveys (aka “bots”) [43]. Therefore, we conducted thorough data cleaning in three stages. First, all responses were briefly reviewed for completeness and obvious random or automated responding. For example, if all open-text and numeric entries contained the number 3 or if all open-text fields contained long, illogical sections of text that appeared to be copied and pasted from websites using an algorithm. All data from these respondents were considered random and were removed from the dataset. At this stage, the respondents were not paid if their data were removed (wave 1 = 367; wave 2 = 61). Second, we reviewed the data more closely to identify inattentive responders. Respondents’ data were excluded entirely if they endorsed starting vaping at less than 8 years of age (wave 1 = 18, wave 2 = 15), reported vaping for more than 20 years (wave 1 = 16, wave 2 = 30), or if they had other inconsistencies that were primarily in their current age versus the age at initial and regular vape use (wave 1 = 17, wave 2 = 1). Bogus and validity check items were also reviewed. Given that the legal age to sell nicotine vapes in the United States is 21, we also excluded data from two respondents from wave 1 who were less than 21 years old. The final dataset for the analysis included 229 respondents from wave 1 and 274 respondents from wave 2, for a total sample of 503 respondents. Some respondents were missing data. and to maintain as many respondents as possible in each analysis, we used listwise deletions. We note the sample size included in each statistical analysis throughout the results.

Several variables were calculated or recoded for analysis. Education was recoded as bachelor’s degree or higher vs. less than a bachelor’s degree based on prior research showing a significant difference in e-cigarette use rates between those with and without a college degree [44]. Employment was categorized as full-time employment vs. not full-time (part-time, unemployed, retired, etc.). Marital status was categorized as married or living with a partner vs. not married (widowed, divorced, never married, etc.). Smoking status was categorized as never—did not smoke 100 cigarettes in lifetime, current—smoked 100 cigarettes in lifetime and currently smoking every day or some days, and former—smoked 100 cigarettes in lifetime and currently smoking not at all. Years of vape use were calculated as current age minus age when regular use began. For the analyses, a natural log transformation was used for the vape times per day variable to manage the positive skew from the outliers. The place of cannabis vape purchase was categorized into store, online vendor, friend/family/individual seller, or other for the regression analyses. Participants who endorsed “Leaf Marijuana” to the question, “What is the primary form of cannabis you use?” were categorized as combustible leaf cannabis users.

### 2.5. Data Analysis

The Statistical Package for the Social Sciences (SPSS v 28.01; [45]) was used for all analyses. We conducted descriptive statistics, one-way analyses of variance for three-group mean comparisons, and independent sample t-tests and chi-square tests for two-group comparisons in sociodemographic variables, vape use behaviors, reasons for vape use, perceived benefits and harms, and measures of dependence. Statistical significance was set at *p* < 0.05 for the univariate analyses. We applied the Bonferroni correction to account for multiple comparisons for the reasons for use (significance at *p* < 0.005) and perceived benefits (significance at *p* < 0.002) and harms (significance at *p* < 0.003).

We conducted two stepwise hierarchical linear regression analyses to identify the correlates of cannabis and nicotine vape dependence and to determine if vape user type (dual vs. single) was associated with vape dependence after controlling for sociodemographics and vape use behaviors. The cannabis vape dependence analysis included dual and cannabis-only vapers, while the nicotine vape dependence analysis included dual and nicotine-only vapers. We only included variables with significant differences across groups in the univariate analyses in the model. The first step included the sociodemographic variables of age, education, employment, marital status, and smoking status (current, former, never). The second step included vape use characteristics—years of vape use and days of vape use in the past 30 days—for both analyses; place of purchase (store, online, friend/individual seller, other) and state medical marijuana card holder vs. not for the cannabis dependence analysis; and nicotine concentration for the nicotine dependence analysis. Due to missing data, stealth vaping was not included as a variable in the nicotine vape dependence regression. Use times per day was not included in the analysis since this is an item in the PSECDI and PSCVDI.

We measured the psychometric properties of the PSCVDI, adapted DSM-V cannabis vape use disorder, and SDS cannabis dependence scales among cannabis vapers (dual and single), including factor analyses, internal consistency, criterion, and convergent validity. We conducted a principal components factor analysis with varimax rotation. Factors with initial eigenvalues >1 were considered meaningful. Cronbach’s alpha was calculated for the total scales and for each factor determined by the factor analysis.

To assess criterion validity, or how well the measure captures dependence or addiction to cannabis, we conducted bivariate correlations between all cannabis measures (PSCVDI, adapted DSM-V, SDS) and with responses to the 0-to-100 VAS scale asking “How addicted are you to cannabis vaping?” We conducted independent t-tests to compare mean dependence scores among those who did and did not endorse using a cannabis vape because of dependence or addiction. Given the discrepancy in group sample sizes, equal variances were not assumed for some t-tests determined by Levene’s Test for Equality of Variance.

To assess convergent validity, or how well each dependence measure corresponds to theoretically related constructs, we conducted bivariate correlations between cannabis dependence measure total scores and cannabis vape use characteristics expected to be associated with dependence, including more years of cannabis vape use, younger age at cannabis vape use initiation, more stealth vaping, and higher THC concentration in e-liquid. We also conducted independent t-tests to assess if the total dependence scores on each measure were significantly higher among co-occurring combustible cigarette (current vs. former/never smokers) and cannabis combustible leaf use and those who had made a failed quit attempt. Given the discrepancy in group sample sizes, equal variances were not assumed for some t-tests determined by Levene’s Test for Equality of Variance.

## 3. Results

### 3.1. Sample Characteristics

Sociodemographic and vape use characteristics are displayed in Table 1. The final sample included 71% dual nicotine and cannabis vapers (*n* = 357), 8% cannabis-only vapers (*n* = 40), and 21% nicotine-only vapers (*n* = 106). The sample had a mean age of 34.1 years (SD = 8.6) and were 62% male, 11% Hispanic (*n* = 55), 88% White (*n* = 439), 8% African American/Black (*n* = 41), 2% Asian (*n* = 11), and 2% other race (*n* = 11). A total of 71% percent of the sample (*n* = 356) had earned a bachelor’s degree or higher, 88% were working full-time (*n* = 441), and the median annual income was $54,999 (IQR = $44,000). Sixty-three percent were married (*n* = 319), 20% were never married (*n* = 99), 12.5% were living with a partner or a member of an unmarried couple (*n* = 63), and 4% were widowed, divorced, or separated (*n* = 20). A total of 83% percent of the sample identified as straight/heterosexual (*n* = 412), 14% as bisexual (*n* = 68), 3% as gay/lesbian/homosexual (*n* = 14), and 1% as other or unwilling to disclose (*n* = 5). A total of 75% percent of the sample were current smokers (*n* = 373), 15% were former smokers, and 10% were never smokers. Compared to cannabis- and nicotine-only users, dual users were younger and more likely to be cigarette smokers, to be married, to be employed full-time, and to hold a bachelor’s degree or higher.

### 3.2. Vape Use Characteristics

Compared to nicotine-only vape users, dual vape users started using their nicotine vape at a younger age, used for more years, used higher nicotine concentrations, reported more frequent stealth vaping, and were more likely to make a prior nicotine vape quit attempt. Dual users were less likely to report using their nicotine vape as a replacement for combustible cigarettes and used their nicotine vape fewer days per week compared to nicotine-only vapers. There was no difference between the dual and nicotine-only vapers in terms of the number of times they used their nicotine vape per day. Dual users were also more likely to purchase their nicotine vape online or from friends or family and less likely to purchase from a gas station. Compared to cannabis-only vapers, dual vapers started using their cannabis vape at a younger age and used for more years and more times per day, but they used for fewer days out of the past 30 days. Dual users were more likely to endorse having a state medical marijuana card but less likely to purchase their cannabis vape from a dispensary. Instead, dual users were more likely to purchase their vape online, from a vape shop or tobacco store, and from friends/family.

### 3.3. Reasons for Vape Use and Perceived Benefits and Harms

Nicotine vape use reasons and perceived benefits and harms are displayed in Table 2. Compared to nicotine-only vape users, dual users were more likely to endorse using their nicotine vape to feel high or alert and because it is new or interesting and were less likely to endorse using their nicotine vape for quitting a combustible product, because they are cheaper than other products, or for dependence or addiction. Dual users were less likely to endorse the benefits of vaping as being less harmful to health than cigarettes, being cheaper than cigarettes, leaving less of an odor on one’s hands, clothes, and breath, tasting good, reducing urges or cravings to smoke, reducing nicotine withdrawal, making it easier to keep from smoking, keeping one from bothering people with smoke, being more socially acceptable, and being healthier than combustible cigarettes compared to nicotine-only vape users. Dual users were less likely to endorse the perceived harms that nicotine vapes contain other chemicals that are not safe and that they still contain nicotine and maintain addiction.

Cannabis vape use reasons and perceived benefits and harms are displayed in Table 3. Dual users were less likely to endorse the ease of use and it being less obvious than smoking a combustible as reasons for cannabis vape use compared to cannabis-only vapers. Dual users were less likely to endorse that cannabis vaping makes their hands and clothes smell less bad and helps them sleep better as perceived benefits and that electronic cigarettes taste bad as a perceived harm compared to cannabis-only vapers. 

### 3.4. Comparisons of Cannabis and Nicotine Vape Dependence by User Type

There were no group differences on the mean cannabis vape dependence on the PSCVDI; however, dual users had significantly higher proportions in the medium and high dependence severity levels compared to cannabis-only users (Table 1). Dual users had significantly higher levels of mean nicotine vape dependence on the PSECDI compared to nicotine-only users; however, group proportions on the dependence severity level on the PSECDI did not differ. Dual users had higher mean scores on the cannabis vape use disorder DSM-V and cannabis dependence SDS scales than cannabis-only users. 

In the stepwise linear regression analyses, younger age and smoking status were significant sociodemographic correlates of higher cannabis vape dependence on the PSCVDI. In a post hoc one-way ANOVA (*n* = 387), current cigarette smokers had higher mean PSCVDI total scores (*M* = 7.53) than former (*M* = 4.85) and never (*M* = 6.23) cigarette smokers [*F*(2386) = 6.40, *p* = 0.002]. Controlling for sociodemographic characteristics, more years of cannabis vape use, more days of cannabis vape use in the past 30 days, and having a state medical cannabis card were associated with higher cannabis vape dependence. In a post hoc one-way ANOVA (*n* = 387), medical marijuana card holders had higher mean PSCVDI total scores (*M* = 9.91) than those without a card (*M* = 5.59) [*F*(1386) = 87.66, *p* < 0.001]. Dual vs. cannabis-only user group was not a significant correlate of cannabis vape dependence after controlling for sociodemographic and vape use characteristics. Younger age was the only sociodemographic variable associated with higher nicotine vape dependence on the PSECDI. Controlling for sociodemographic characteristics, more years of vape use, more days of vape use in the past 30 days, and lower nicotine concentration were associated with higher nicotine vaping dependence. Dual vs. nicotine-only user group was not a significant correlate of nicotine vape dependence after controlling for sociodemographic variables and vape use behaviors. The results are displayed in Table 4.

### 3.5. Psychometric Assessment of Cannabis Vape Measures

The PSCVDI revealed a two-factor structure, with the first two items comprising one factor that explained 13.8% of the variance and the remaining seven items explaining 49.6%. The internal consistency of the PSCVDI total score was adequate (Cronbach’s alpha = 0.61; *n* = 379), the consistency of the factor containing items 1 and 2 was Cronbach’s alpha = 0.44 (*n* = 382), and the factor containing the remaining seven items was Cronbach’s alpha = 0.85 (*n* = 386). The DSM-V cannabis vape use disorder scale revealed a single factor explaining 57.3% of the variance with an internal consistency of Cronbach’s alpha = 0.92 (*n* = 365). The SDS cannabis dependence scale revealed a single factor explaining 64.4% of the variance with an internal consistency of Cronbach’s alpha = 0.86 (*n* = 373). 

For the assessment of criterion validity, all dependence measures were significantly, positively correlated with one another when calculated among all cannabis vapers (Table 5). All dependence measures were also significantly correlated with the VAS item, “How addicted are you to cannabis vaping?” The mean DSM-V scores were significantly higher for those who endorsed that their primary reason for using a cannabis vape was dependence or addiction (*M* = 6.58, *SD* = 3.69, *n* = 24) compared to those who did not endorse dependence or addiction as a reason for use [*M* = 3.64, *SD* = 3.50, *n* = 364; *t*(386) = 3.98, *p* < 0.001]. Mean SDS scores were significantly higher for those who endorsed that their primary reason for using a cannabis vape was dependence or addiction (*M* = 6.92, *SD* = 3.41, *n* = 24) compared to those who did not endorse dependence or addiction as a reason for use [*M* = 3.78, *SD* = 3.21, *n* = 368; *t*(390) = 4.63, *p* < 0.001]. PSCVDI scores were not significantly different among those who did and did not endorse dependence or addiction as a reason for use [*t*(388) = 1.75, *p* = 0.081). 

For the assessment of convergent validity, all measures were significantly correlated with years of cannabis vape use, age at initiation, stealth vaping, and e-liquid THC percentage, although the PSCVDI had lower correlations than the DSM-V and SDS measures (Table 5). Mean PCVDI scores were significantly higher among current cigarette smokers (*M* = 7.53, *SD* = 5.02, *n* = 304) vs. former/never smokers [*M* = 5.71, *SD* = 3.75, *n* = 86; *t*(180.17) = 3.66, *p* < 0.001] and among those who had made a failed cannabis vape quit attempt (*M* = 10.43, *SD* = 5.33, *n* = 95) vs. those who had not [*M* = 6.09, *SD* = 4.07, *n* = 288; *t*(131.90) = 7.26, *p* < 0.001]. Mean DSM-V scores were significantly higher among current cigarette smokers (*M* = 4.21, *SD* = 3.71, *n* = 302) vs. former/never smokers [*M* = 2.47, *SD* = 2.66, *n* = 86; *t*(188.97) = 4.88, *p* < 0.001] and among those who had made a failed cannabis vape quit attempt (*M* = 6.97, *SD* = 3.64, *n* = 95) vs. those who had not [*M* = 2.82, *SD* = 2.89, *n* = 286; *t*(135.59) = 10.11, *p* < 0.001]. Mean SDS scores were significantly higher among current cigarette smokers (*M* = 4.44, *SD* = 3.24, *n* = 305) vs. former/never smokers [*M* = 2.34, *SD* = 3.00, *n* = 87; t(390) = 5.40, *p* < 0.001] and among those who had made a failed cannabis vape quit attempt (*M* = 5.96, *SD* = 2.55, *n* = 98) vs. those who had not [*M* = 3.36, *SD* = 3.27, *n* = 287; *t*(213.40) = 8.08, *p* < 0.001]. There were no significant differences between combustible cannabis users vs. non-combustible cannabis users for all dependence scores.

## 4. Discussion

This cross-sectional survey adds to the current literature on cannabis and nicotine vaping by comparing dual nicotine and cannabis vape users to those who vape nicotine and cannabis alone. Whereas dual and cannabis-only users reported similar cannabis vape use motivations, behaviors, and dependence levels, dual and nicotine-only users reported differences in nicotine vape use and combustible cigarette use that may lead to more nicotine dependence or physical harm for dual users. In the current sample, dual users were younger and started using nicotine and cannabis vapes at a younger age than single users, with the average age of regular use beginning in their mid-20s compared to the early 30s for the single users. We know from prior research that an early-onset of cannabis use is linked with a higher risk of addiction and polysubstance use [46]. Therefore, it may not be surprising that the dual users in our sample were also more likely to be cigarette smokers. 

In our sample, dual users were more likely to be married or living with a partner and be college-educated. This is not consistent with a national survey of adult dual users of combustible tobacco/cannabis, showing higher proportions among those who have never been married and no differences in education compared to combustible tobacco or cannabis-only users [47]. Although MTurk samples typically include adults with higher educational attainment than the U.S. population [43], our findings may point to the unique demographics of vape users compared to combustible nicotine and cannabis users. National surveys show that nicotine e-cigarette users are more likely to be younger, non-Hispanic, and White compared to combustible cigarette users [48]. Unlike rates of cigarette use, which go down with increasing education levels, rates of e-cigarette use increase with education level up to some college education and then drop to the lowest levels for those with a college degree [44]. To our knowledge, our study is the first to show that dual nicotine/cannabis vape users may be more likely to have a college degree than single-substance vape users.

Dual users vaped cannabis fewer days out of the past 30 days but more times per day, on average, compared to the cannabis-only vapers. This pattern may be indicative of using cannabis vapes for recreational purposes to get high; however, dual users were not more likely to endorse getting high as a primary reason for use compared to cannabis-only vape users. Among all cannabis vapers in the survey, about a quarter endorsed feeling high or alert as reasons for use, and a third endorsed the taste, satisfaction, and ease of use as primary reasons. These reasons were echoed in the perceived benefits of cannabis vape use; about a quarter of cannabis vapers endorsed the convenience of not having to go outside and that it tastes good. This suggests that cannabis vape users are largely using these devices for recreation and enjoyment rather than for health benefits such as harm reduction compared to combustible cannabis or to manage medical or psychiatric symptoms.

In contrast, clear differences between dual users and nicotine-only vapers emerged as reasons for use and perceived benefits and harms of nicotine vapes. Nicotine-only vapers primarily reported vaping to quit combustible cigarettes and endorsed the health, financial, and social benefits of vapes over combustible cigarettes. Previous studies have shown nicotine vapes to be a useful harm-reduction tool; they are more efficacious when used as a complete substitute for combustible cigarettes and in preventing relapse [49,50,51]. A third of nicotine-only vapers were former smokers, and 76% reported that they were using their nicotine vape to replace combustible cigarettes compared to only 66% of dual users. Dual users, on the other hand, were more likely to endorse that they used their nicotine vape to feel high or alert and because it was new or interesting. Similar to cannabis vape use, these results suggest that dual users may be more likely to use their nicotine vape for recreation as opposed to harm-reduction. Prior research has found that dual users may use nicotine vapes to enhance the high from cannabis vapes, which may be one explanation of our findings [19]. Dual users who use their nicotine vape for recreation and to feel high or alert rather than to replace combustible cigarettes may also be smoking more combustible cigarettes in addition to using a vape. Further, a higher proportion of dual users were combustible cigarette smokers compared to nicotine-only vape users. Considering that combustible cigarettes lead to death in half of all users [52], this finding may be among the most concerning for dual nicotine and cannabis vape users in the current sample.

Our results do not suggest that dual cannabis/nicotine vapers are more dependent on their cannabis vapes compared to cannabis-only users. Rather, being a current combustible cigarette smoker, being younger, being married, using a cannabis vape for more years, and holding a medical marijuana card with one’s state were all related to higher cannabis vaping dependence scores. The results were similar for nicotine dependence, with sociodemographic and vape use characteristics accounting for group differences in nicotine vape dependence rather than dual user status. These findings suggest that vaping cannabis and nicotine may not be a risk for higher vaping dependence on either substance. Rather, younger vape users who have used for longer and in combination with combustible cigarettes may benefit the most from intervention. Unexpectedly, we found that lower nicotine concentration e-liquids were associated with higher nicotine vape dependence after controlling for sociodemographic characteristics. Although statistically significant, the effect size of the association was small and may be due to variations in how respondents determined their nicotine strength. Further research on the role of nicotine concentration in vape dependence is warranted, especially in consideration of other factors that influence nicotine exposure from e-cigarettes, including the power of the device and the e-liquid formulation [53].

To our knowledge, this study is the first to assess the psychometric properties of assessments for cannabis vaping dependence specifically. We found adequate psychometric properties for an adapted version of the PSECDI, where the word “e-cigarette” was changed to “cannabis vape” (PSCVDI), and for the DSM-V cannabis use disorder criteria, with the word “cannabis” changed to “cannabis vape”. The PSCVDI had good criterion and convergent validity with other measures of cannabis dependence/addiction and vape use characteristics, polysubstance use, and quit attempts. Similar to previous studies on the PSECDI, the PSCVDI did not have a single factor structure [54]. The first two items, cannabis vape use times per day and time to use after waking, were separated into their own factor. When inquiring about cigarette dependence, these items comprise the Heaviness of Smoking Index, a measure highly predictive of quitting maintenance [38]. These items, however, did not have good internal consistency among the current sample and may not hold the same relevance to dependence for cannabis as they do for nicotine. There is limited research on the predictive validity of the time to cannabis use after waking. A survey of German college students found that only 1–2% endorsed using cannabis upon waking most of the time, but these individuals were more likely to meet the criteria for DSM-IV cannabis dependence [55]. No studies, to our knowledge, have assessed cannabis vaping upon waking specifically, and future studies could help determine its relevance for health outcomes. Future studies focused solely on the development of a validated cannabis vaping dependence assessment tool would also benefit from a larger sample that included participants who exclusively vape cannabis to avoid the confounding effects of combustible cannabis use. The current study shows that validated e-cigarette and cannabis dependence assessments are adequate substitutes for measuring cannabis vape dependence until more validation studies are completed. 

### Limitations

This study had a number of limitations, including a convenience sample and several challenges with the survey development. The sample was primarily male, White, and college-educated. This is in part consistent with U.S. statistics showing a high prevalence of White males among nicotine e-cigarette users; however, the rates of use are lowest among those that are college-educated [44]. Samples collected via MTurk are convenience samples and are not nationally representative; therefore, they may not generalize to the demographic groups most likely to vape. Alternatively, MTurk is an ideal platform to recruit e-cigarette and vape users, since workers tend to over-represent substance users, including cannabis users [30]. As with any survey study, we were not able to biochemically verify that participants were nicotine or cannabis users. We purposely masked the true eligibility criteria for the study, and participants had little incentive to lie about their use given the many work opportunities on MTurk and the modest compensation provided for completing the current study. The anonymity of MTurk facilitates honest and unbiased responding when assessing stigmatized behaviors, which cannot always be achieved in face-to-face laboratory studies [56,57]. We also employed a number of data validity and reliability checks in the current survey given that prior studies have found responses from cannabis users to be less reliable than those from alcohol users [57].

Despite our attempt to provide definitions in the beginning of the survey, we have some concerns regarding how participants interpreted the terminology we used in our survey questions, especially for dual users that were asked the same questions about cannabis and nicotine vape use. In an attempt to not alter survey items from standardized and previously used measures, we left the term “electronic cigarette” in some surveys and changed it to “vape” in others. In addition, it is not clear that all participants understood what we meant by “cannabis”. Some participants who endorsed being cannabis vapers did not endorse being a general cannabis user later in the survey. We therefore caution readers to interpret our results of “combustible leaf cannabis users” with that in mind, since some leaf users may have been missed among those who did not endorse cannabis use. Unfortunately, this is a common problem for those researching new and emerging trends in substance use, since the terms can be regional and change rapidly in the commercial and underground markets. For future research, we suggest re-defining the terms at several points throughout the survey, being careful not to branch survey items or tools based on questions with potentially confusing terminology, and conducting preliminary research on the common names for substances among the population of interest to inform survey development. For example, some researchers have piloted innovative ways to identify emerging cannabis terms using social media [58].

## 5. Conclusions

The current survey adds to the scant literature on dual nicotine and cannabis vaping by comparing sociodemographic and vape use characteristics, reasons for use, perceived benefits and harms of use, and cannabis and nicotine dependence among dual and single-substance vape users. While we did not find many differences between dual vape users and cannabis-only vape users, nicotine-only users primarily used their nicotine vape for harm-reduction as a smoking alternative, while dual users did not. Somewhat paradoxically, dual users are more likely to be combustible cigarette smokers and, therefore, are at a high risk for cigarette-related harm from toxicant exposure. After controlling for relevant use variables, such as years of use and past 30-day use frequency, dual users were not more likely to be dependent on vaping compared to single users. The current study provides support for the use of adapted versions of the PSECDI and DSM-V criteria to measure cannabis vaping dependence specifically, although future research among exclusive cannabis vapers is warranted.

## Figures and Tables

**Table 1 ijerph-19-10392-t001:** Sociodemographic, nicotine vape use, and cannabis vape use characteristics by vape user type.

	Nicotine-Only (*n* = 106)	Dual Nicotine and Cannabis (*n* = 357)	Cannabis-Only (*n* = 40)	Total (*N* = 503)	*p*-Value
Sociodemographic Characteristics					
Age, *M* (*SD*) (*n* = 503)	37.26 (9.49)	32.83 (7.91)	37.25 (9.01)	34.11 (8.57)	*F* (2, 502) = 14.61, *p* < 0.001
Gender Identity, *n* (%) (*n* = 501)					*χ^2^*(6) = 5.98, *p* = 0.425
Male	65 (61.3)	225 (63.2)	21 (53.8)	311 (62.1)	
Female	40 (37.7)	129 (36.2)	18 (46.2)	187 (37.3)	
Transmale	1 (0.9)	0 (0)	0 (0)	1 (0.2)	
Unknown	0 (0)	2 (0.6)	0 (0)	2 (0.4)	
Race, *n* (%) (*n* = 502)					*χ^2^*(8) = 11.54, *p* = 0.173
White	93 (88.6)	312 (87.4)	34 (85.0)	439 (87.5)	
Black/African American	7 (6.7)	32 (9.0)	2 (5.0)	41 (8.2)	
Asian	5 (4.8)	4 (1.1)	2 (5.0)	11 (2.2)	
Other	0 (0)	9 (2.5)	2 (5.0)	11 (2.2)	
Hispanic/Latino, *n* (%) (*n* = 503)	7 (6.6)	46 (12.9)	2 (5.0)	55 (10.9)	*χ^2^*(2) = 4.88, *p* = 0.087
Married or Living with Partner, *n* (%) (*n* = 503)	66 (62.3)	280 (78.4)	26 (65.0)	372 (74.0)	*χ^2^*(2) =12.90, *p* = 0.002
Sexual Orientation, *n* (%) (*n* = 499)					*χ^2^*(10) = 10.40, *p* = 0.406
Heterosexual	91 (85.8)	287 (81.1)	34 (87.2)	412 (82.6)	
Homosexual (Gay)	2 (1.9)	5 (1.4)	1 (2.6)	8 (1.6)	
Homosexual (Lesbian)	3 (2.8)	2 (0.6)	1 (2.6)	6 (1.2)	
Bisexual	9 (8.5)	56 (15.8)	3 (7.7)	68 (13.6)	
Other or undisclosed	1 (0.9)	4 (1.2)	0 (0)	5 (1.0)	
Bachelor’s degree or higher, *n* (%) (*n* = 501)	57 (53.8)	275 (77.5)	24 (60.0)	356 (71.1)	*χ^2^*(2) = 24.86, *p* < 0.001
Current full-time employment, *n* (%) (*n* = 503)	91 (85.8)	322 (90.2)	28 (70.0)	441 (87.7)	*χ^2^*(2) = 13.99, *p* < 0.001
Household income, Med (IQR) (*n* = 493)	54,999 (44,000)	54,999 (39,000)	64,999 (51,000)	54,999 (44,000)	*χ^2^*(2) = 1.89, *p* = 0.390
Cigarette smoking status, *n* (%) (*n* = 499)					*χ^2^*(4) = 95.32, *p* < 0.001
Current	64 (61.0)	297 (83.7)	12 (30.8)	373 (74.7)	
Former	35 (33.3)	29 (8.2)	11 (28.2)	75 (15.0)	
Never	6 (5.7)	29 (8.2)	16 (41.0)	51 (10.2)	
Nicotine Vape Use Characteristics					
Age of regular use, *M* (*SD*) (*n* = 456)	33.01 (10.23)	26.19 (7.68)		27.78 (8.82)	*t*(142.68) = 6.34, *p* < 0.001
Years of use, *M* (*SD*) (*n* = 456)	4.25 (3.44)	6.66 (4.49)		6.10 (4.39	*t*(223.79) = 5.85, *p* < 0.001
Days of use in past 30 days, *M* (*SD*) (*n* = 437)	20.73 (10.49)	16.06 (9.82)		17.16 (10.16)	*t*(435) = 4.15, *p* < 0.001
Times per day, Med (IQR) (n = 460)	5 (12)	6 (12)		6 (12)	*χ^2^*(1) = 0.16, *p* = 0.740
Nicotine concentration, *M* (*SD*) (*n* = 422)	14.39 (23.36)	24.81 (73.21)		22.24 (64.72)	*t*(419.86) = 2.22, *p* = 0.027
Stealth vaping, *n* (%) (*n* = 260)					*χ^2^*(4) = 27.85, *p* < 0.001
Never	21 (36.8)	24 (11.8)		45 (17.3)	
Rarely	21 (36.8)	57 (28.1)		78 (30.0)	
A few times a month	10 (17.5)	67 (33.0)		77 (29.6)	
A few times a week	5 (8.8)	44 (21.7)		49 (18.8)	
Daily	0 (0.0)	11 (5.4)		11 (4.2)	
Place of purchase (*n* = 456)					*χ^2^*(5) = 26.16, *p* < 0.001
Gas station	15 (14.2)	26 (7.4)		41 (9.0)	
Tobacco store or vape shop	50 (47.2)	161 (46.0)		211 (46.3)	
Online store	34 (32.1)	116 (33.1)		150 (32.9)	
Online marketplace	3 (2.8)	31 (8.9)		34 (7.5)	
Friends/family	0 (0.0)	16 (4.6)		16 (3.5)	
Use of vape to replace combustible, *n* (%) (*n* = 457)	80 (76.2)	232 (65.9)		312 (68.3)	*χ^2^*(1) = 3.95, *p* = 0.047
Lifetime quit attempt, *n* (%) (*n* = 460)	18 (17.0)	124 (35.0)		142 (30.9)	*χ^2^*(1) = 12.45, *p* < 0.001
Cannabis Vape Use Characteristics					
Age of regular use, *M* (*SD*) (*n* = 380)		27.08 (7.91)	33.90 (9.91)	27.80 (8.39)	*t*(45.03) = 4.20, *p* < 0.001
Years of use, *M* (*SD*), *(n* = 380)		5.87 (4.83)	3.35 (3.33)	5.61 (4.75)	*t*(60.19) = 4.28, *p* < 0.001
Days of use in past 30 days, *M* (*SD*) (*n* = 385)		7.61 (8.55)	13.20 (9.46)	8.19 (8.80)	*t*(383) = 3.87, *p* < 0.001
Times per day, Med (IQR) (*n* = 382)		4 (9.5)	1 (1.0)	3 (9)	*χ^2^*(1) = 17.93, *p* < 0.001
THC percentage, *M* (*SD*) (*n* = 385)		55.62 (24.11)	56.28 (28.50)	55.69 (24.55)	*t*(44.35) = 0.14, *p* = 0.890
Stealth vaping, *n* (%) (*n* = 391)					*χ^2^*(4) = 7.31, *p* = 0.120
Never		85 (24.2)	16 (40.0)	101 (25.8)	
Rarely		126 (35.9)	13 (32.5)	139 (35.5)	
A few times a month		91 (25.9)	10 (25.0)	101 (25.8)	
A few times a week		43 (12.3)	1 (2.5)	44 (11.3)	
Daily		6 (1.7)	0 (0.0)	6 (1.5)	
Place of purchase, *n* (%) (*n* = 387)					*χ^2^*(7) = 74.20, *p* < 0.001
Gas station		8 (2.3)	0 (0.0)	8 (2.1)	
Tobacco store or vape shop		102 (29.4)	1 (2.5)	103 (26.6)	
Online store		69 (19.9)	3 (7.5)	72 (18.6)	
Online marketplace		39 (11.2)	1 (2.5)	40 (10.3)	
Friends/family		54 (15.6)	4 (10.0)	58 (15.0)	
Individual Seller		36 (10.4)	8 (20.0)	44 (11.4)	
Dispensary		35 (10.1)	23 (57.5)	58 (15.0)	
Other		4 (1.2)	0 (0)	4 (1.0)	
Use of vape to replace combustible, *n* (%) (*n* = 386)		155 (44.8)	19 (47.5)	174 (45.1)	*χ^2^*(1) = 0.106, *p* = 0.745
Lifetime quit attempt, *n* (%) (*n* = 389)		93 (26.6)	5 (12.8)	98 (25.2)	*χ^2^*(1) = 3.52, *p* = 0.061
Combustible (leaf) cannabis use, *n* (%) (*n* = 503)	33 (31.1)	135 (37.8)	21 (52.5)	189 (37.6)	*χ^2^*(2) = 5.68, *p* = 0.058
State medical marijuana card holder, *n* (%) (*n* = 430)	3 (8.1)	137 (38.7)	7 (17.9)	147 (34.2)	*χ^2^*(2) = 18.96, *p* < 0.001
Dependence Measures					
PSECDI Total Score (*n* = 462)	8.03 (3.46)	9.13 (4.44)		8.87 (4.26)	*F*(1, 460) = 5.49, *p* = 0.020
PSECDI Dependence Level (*n* = 462)					*χ^2^*(3) = 5.33, *p* = 0.149
Not dependent	9 (8.5)	32 (9.0)		41 (8.9)	
Low dependence	48 (45.3)	138 (38.8)		186 (40.3)	
Medium dependence	36 (34.0)	107 (30.1)		143 (31.0)	
High dependence	13 (12.3)	79 (22.2)		92 (19.9)	
PSCVDI Total Score (*n* = 390)		7.27 (4.98)	5.85 (2.88)	7.13 (4.8)	*F* (1, 388) = 3.13, *p* = 0.078
PSCVDI Dependence Level (*n* = 390)					*χ^2^*(3) = 11.75, *p* = 0.008
Not dependent		80 (22.9)	6 (15.0)	86 (22.1)	
Low dependence		139 (39.7)	27 (67.5)	166 (42.6)	
Medium dependence		78 (22.3)	5 (12.5)	83 (21.3)	
High dependence		53 (15.1)	2 (5.0)	55 (14.1)	
Cannabis Dependence Severity (SDS) (*n* = 392)		4.26 (3.28)	1.43 (2.24)	3.97 (3.30)	*F* (1390) = 28.36, *p* < 0.001
Cannabis Vape Dependence (DSM) (*n* = 388)		4.07 (3.66)	1.63 (1.46)	3.82 (3.58)	*F*(1386) = 17.55, *p* < 0.001

**Table 2 ijerph-19-10392-t002:** Reasons for use and perceived benefits and harms of nicotine vape use by vape user type.

	Nicotine-Only (*n* = 106)	Dual Nicotine and Cannabis (*n* = 357)	Total (*n* = 463)	*p*-Value
	*n* (%)	*n* (%)	*n* (%)	
Nicotine Vape Reasons for Use				
Ease of use	57 (53.8)	180 (50.4)	237 (51.2)	*χ^2^*(1) = 0.37, *p* = 0.544
Quitting a combustible product	52 (49.1)	65 (18.2)	117 (25.3)	***χ^2^*(1) = 41.19, *p* < 0.001**
Health benefits	26 (24.5)	93 (26.1)	119 (25.7)	*χ^2^*(1) = 0.10, *p* = 0.753
Taste	42 (39.6)	140 (39.2)	182 (39.3)	*χ^2^*(1) = 0.01, *p* = 0.940
Feeling (high or alertness)	6 (5.7)	94 (26.3)	100 (21.6)	***χ^2^*(1) = 20.62, *p* < 0.001**
Satisfaction	39 (36.8)	148 (41.5)	187 (40.4)	*χ^2^*(1) = 0.74, *p* = 0.390
Less obvious than smoking a combustible	26 (24.5)	68 (19.0)	94 (20.3)	*χ^2^*(1) = 1.52, *p* = 0.218
New and interesting	10 (9.4)	91 (25.5)	101 (21.8)	***χ^2^*(1) = 12.35, *p* < 0.001**
Cheaper than other products	29 (27.4)	55 (15.4)	84 (18.1)	***χ^2^*(1) = 7.86, *p* = 0.005**
Dependence or addiction	28 (26.4)	41 (11.5)	69 (14.9)	***χ^2^*(1) = 14.37, *p* < 0.001**
To lose weight	2 (1.9)	8 (2.2)	10 (2.2)	*χ^2^*(1) = 0.05, *p* = 0.826
Perceived Benefits of Nicotine Vape Use				
Helps me cut down on the number of cigarettes I smoke	56 (52.8)	150 (42.0)	206 (44.5)	*χ^2^*(1) = 3.87, *p* = 0.049
Helped me quit smoking	43 (40.6)	111 (31.1)	154 (33.3)	*χ^2^* (1)= 3.31, *p* = 0.069
Good to use in places where cigarette smoking is not allowed	39 (36.8)	107 (30.0)	146 (31.5)	*χ^2^*(1) = 1.76, *p* = 0.184
Makes it so I do not have to go outside to smoke	42 (39.6)	94 (26.3)	136 (29.4)	*χ^2^*(1) = 6.96, *p* = 0.008
Less harmful to my health than smoking cigarettes	56 (52.8)	119 (33.3)	175 (37.8)	***χ^2^*(1) = 13.22, *p* < 0.001**
Reduces harmful effects on my family or friends	35 (33.0)	82 (23.0)	117 (25.3)	*χ^2^*(1) = 4.37, *p* = 0.037
Cheaper than smoking cigarettes	45 (42.5)	81 (22.7)	126 (27.2)	***χ^2^*(1) = 16.12, *p* < 0.001**
Reduces the risk of lung cancer	27 (25.5)	81 (22.7)	108 (23.3)	*χ^2^*(1) = 0.35, *p* = 0.552
Reduces the risk of mouth or throat cancer	19 (17.9)	69 (19.3)	88 (19.0)	*χ^2^*(1) = 0.11, *p* = 0.746
Reduces my coughing with mucous	27 (25.5)	52 (14.6)	79 (17.1)	*χ^2^*(1) = 6.87, *p* = 0.009
Makes me less short of breath and improves my breathing	24 (22.6)	47 (13.2)	71 (15.3)	*χ^2^*(1) = 5.65, *p* = 0.017
Makes my hands and clothes smell less bad	53 (50.0)	66 (18.5)	119 (25.7)	***χ^2^*(1) = 42.50, *p* < 0.001**
Reduces my bad breath and bad odors	39 (36.8)	58 (16.2)	97 (21.0)	***χ^2^*(1) = 20.83, *p* < 0.001**
Improves my sense of smell and ability to taste	20 (18.9)	44 (12.3)	64 (13.8)	*χ^2^*(1) = 2.94, *p* = 0.087
Tastes good	57 (53.8)	119 (33.3)	176 (38.0)	***χ^2^*(1) = 14.49, *p* < 0.001**
Feels good when inhaling	36 (34.0)	110 (30.8)	146 (31.5)	*χ^2^*(1) = 0.38, *p* = 0.540
Reduces my urges or craving to smoke	50 (47.2)	57 (16.0)	107 (23.1)	***χ^2^*(1) = 44.78, *p* < 0.001**
Makes it easier to keep from smoking cigarettes	52 (49.1)	58 (16.2)	110 (23.8)	***χ^2^*(1) = 48.57, *p* < 0.001**
Keeps me from bothering other people with my smoke	35 (33.0)	50 (14.0)	85 (18.4)	***χ^2^*(1) = 19.71, *p* < 0.001**
Takes away my craving to smoke faster than cigarette smoking does	11 (10.4)	30 (8.4)	41 (8.9)	*χ^2^*(1) = 0.40, *p* = 0.530
Gives me as much or more nicotine than I can get by smoking cigarettes	18 (17.0)	27 (7.6)	45 (9.7)	*χ^2^*(1) = 8.26, *p* = 0.004
Reduces nicotine withdrawal	28 (26.4)	49 (13.7)	77 (16.6)	***χ^2^*(1) = 9.49, *p* = 0.002**
Helps me sleep better	9 (8.5)	42 (11.8)	51 (11.0)	*χ^2^*(1) = 0.89, *p* = 0.344
Produces large clouds and shapes from vapor	8 (7.5)	36 (10.1)	44 (9.5)	*χ^2^*(1) = 0.61, *p* = 0.434
Trendy	11 (10.4)	65 (18.2)	76 (16.4)	*χ^2^*(1) = 3.65, *p* = 0.056
More socially acceptable than smoking	33 (31.3)	52 (14.6)	85 (18.4)	***χ^2^*(1) = 14.97, *p* < 0.001**
Allows me to experiment with flavors	19 (17.9)	44 (12.3)	63 (13.6)	*χ^2^*(1) = 2.18, *p* = 0.140
Gets me high	1 (0.9)	27 (7.6)	28 (6.0)	*χ^2^*(1) = 6.30, *p* = 0.012
Makes me more alert	10 (9.4)	31 (8.7)	41 (8.9)	*χ^2^*(1) = 0.057, *p* = 0.811
Healthier than combustible cigarettes	34 (32.1)	38 (10.6)	72 (15.6)	***χ^2^*(1) = 28.59, *p* < 0.001**
Perceived Harms of Nicotine Vape Use				
The vapor that they make contains other chemicals which are not safe; they can hurt my health	45 (42.5)	84 (23.5)	129 (27.9)	***χ^2^*(1) = 14.56, *p* < 0.001**
I am still getting nicotine so I would stay addicted	53 (50.0)	110 (30.8)	163 (35.2)	***χ^2^*(1) = 13.19, *p* < 0.001**
The nicotine vapor from electronic cigarettes causes lung cancer	12 (11.3)	64 (17.9)	76 (16.4)	*χ^2^*(1) = 2.60, *p* = 0.107
The nicotine vapor from electronic cigarettes causes wet cough with mucous (brown liquid)	9 (8.5)	68 (19.0)	77 (16.6)	*χ^2^*(1) = 6.57, *p* = 0.010
The nicotine vapor from electronic cigarettes causes a dry cough	10 (9.4)	68 (19.0)	78 (16.8)	*χ^2^*(1) = 5.39, *p* = 0.020
The nicotine vapor from electronic cigarettes causes lung problems	22 (20.8)	77 (21.6)	99 (21.4)	*χ^2^*(1) = 0.03, *p* = 0.858
The nicotine vapor from electronic cigarettes causes mouth or throat cancer	10 (9.4)	58 (16.2)	68 (14.7)	*χ^2^*(1) = 3.03, *p* = 0.082
The nicotine vapor from electronic cigarettes makes me short of breath	4 (3.8)	43 (12.0)	47 (10.2)	*χ^2^*(1) = 6.13, *p* = 0.013
Using them burns my throat.	12 (11.3)	65 (18.2)	77 (16.6)	*χ^2^*(1) = 2.80, *p* = 0.095
Using them gives me a dry mouth or dry throat	18 (17.0)	67 (18.8)	85 (18.4)	*χ^2^*(1) = 0.17, *p* = 0.677
Electronic cigarettes have toxic substances in them	25 (23.6)	71 (19.9)	96 (20.7)	*χ^2^*(1) = 0.68, *p* = 0.410
Electronic cigarettes do not stop me from having urges or cravings to smoke	16 (15.1)	41 (11.5)	57 (12.3)	*χ^2^*(1) = 0.99, *p* = 0.321
I do not get enough nicotine from electronic cigarettes	7 (6.6)	40 (11.2)	47 (10.2)	*χ^2^*(1) = 1.90, *p* = 0.168
It is too difficult to adjust how much nicotine I get with electronic cigarettes	7 (6.6)	36 (10.1)	43 (9.3)	*χ^2^*(1) = 1.18, *p* = 0.278
Electronic cigarettes are addicting	36 (34.0)	79 (22.1)	115 (24.8)	*χ^2^*(1) = 6.13, *p* = 0.013
Electronic cigarettes do not help me quit smoking	17 (16.0)	52 (14.6)	69 (14.9)	*χ^2^*(1) = 0.14, *p* = 0.709
If I use electronic cigarettes to quit smoking cigarettes, I will just go back to smoking when I stop using them	25 (23.6)	51 (14.3)	76 (16.4)	*χ^2^*(1) = 5.15, *p* = 0.023
Using electronic cigarettes gives me headaches or nausea or makes me feel dizzy	4 (3.8)	34 (9.5)	38 (8.2)	*χ^2^*(1) = 3.59, *p* = 0.058
Electronic cigarettes taste bad	2 (1.9)	14 (3.9)	16 (3.5)	*χ^2^*(1) = 1.01, *p* = 0.314
Electronic cigarettes cause me to gain weight	2 (1.9)	12 (3.4)	14 (3.0)	*χ^2^*(1) = 0.61, *p* = 0.436

Note: With the Bonferroni correction, statistical significance was set at *p* < 0.005 for reasons for use, *p* < 0.002 for perceived benefits, and *p* < 0.003 for perceived harms. Bolded statistics indicate statistical significance.

**Table 3 ijerph-19-10392-t003:** Reasons for and concerns about cannabis vape use by vape user type.

	Dual Nicotine and Cannabis (*n* = 357)	Cannabis-Only (*n* = 40)	Total (*n* = 397)	*p*-Value
	*n* (%)	*n* (%)	*n* (%)	
Cannabis Vape Reasons for Use				
Ease of use	126 (35.3)	26 (65.0)	152 (38.3)	***χ^2^*(1) = 13.43, *p* < 0.001**
Quitting a combustible product	38 (10.6)	3 (7.5)	41 (10.3)	*χ^2^*(1) = 0.38, *p* = 0.535
Health benefits	82 (23.0)	6 (15.0)	88 (22.2)	*χ^2^*(1) = 1.32, *p* = 0.250
Taste	117 (32.8)	8 (20.0)	125 (31.5)	*χ^2^*(1) = 2.72, *p* = 0.099
Feeling (high or alertness)	91 (25.5)	17 (42.5)	108 (27.2)	*χ^2^*(1) = 5.26, *p* = 0.022
Satisfaction	122 (34.2)	10 (25.0)	132 (33.2)	*χ^2^*(1) = 1.36, *p* = 0.243
Less obvious than smoking a combustible	60 (16.8)	14 (35.0)	74 (18.6)	***χ^2^*(1) = 7.85, *p* = 0.005**
New and interesting	59 (16.5)	6 (15.0)	65 (16.4)	*χ^2^*(1) = 0.06, *p* = 0.805
Cheaper than other products	21 (5.9)	0 (0.0)	21 (5.3)	*χ^2^*(1) = 2.48, *p* = 0.115
Dependence or addiction	23 (6.4)	1 (2.5)	24 (6.0)	*χ^2^*(1) =0.98, *p* = 0.321
To lose weight	8 (2.2)	0 (0.0)	8 (2.0)	*χ^2^*(1) = 0.92, *p* = 0.339
Perceived Benefits of Cannabis Vape Use				
Helps me cut down on the amount of marijuana I smoke	59 (16.5)	7 (17.5)	66 (16.6)	*χ^2^*(1) = 0.03, *p* = 0.875
Helped me quit smoking marijuana	66 (18.5)	1 (2.5)	67 (16.9)	*χ^2^*(1) = 6.55, *p* = 0.010
Is good to use in places where smoking marijuana is not allowed	81 (22.7)	12 (30.0)	93 (23.4)	*χ^2^*(1) = 1.07, *p* = 0.301
Makes it so I do not have to go outside to smoke	89 (24.9)	11 (27.5)	100 (25.2)	*χ^2^*(1) = 0.13, *p* = 0.723
Is less harmful to my health than smoking marijuana	69 (19.3)	7 (17.5)	76 (19.1)	*χ^2^*(1) = 0.08, *p* = 0.781
Reduces harmful effects on my family or friends	62 (17.4)	2 (5.0)	64 (16.1)	*χ^2^*(1) = 4.07, *p* = 0.044
Is cheaper than smoking marijuana	55 (15.4)	3 (7.5)	58 (14.6)	*χ^2^*(1) = 1.80, *p* = 0.179
Reduces the risk of lung cancer	52 (14.6)	5 (12.5)	57 (14.4)	*χ^2^*(1) = 0.13, *p* = 0.724
Reduces the risk of mouth or throat cancer	37 (10.4)	1 (2.5)	38 (9.6)	*χ^2^*(1) = 2.57, *p* = 0.109
Reduces my coughing with mucous	38 (10.6)	7 (17.5)	45 (11.3)	*χ^2^*(1) = 1.68, *p* = 0.195
Makes me less short of breath and improves my breathing	32 (9.0)	5 (12.5)	37 (9.3)	*χ^2^*(1) = 0.53, *p* = 0.466
Makes my hands and clothes smell less bad	44 (12.3)	14 (35.0)	58 (14.6)	***χ^2^*(1) = 14.83, *p*= <0.001**
Reduces my bad breath and bad odors	40 (11.2)	3 (7.5)	43 (10.8)	*χ^2^*(1) = 0.51, *p* = 0.475
Improves my sense of smell and ability to taste	32 (9.0)	0 (0.0)	32 (8.1)	*χ^2^*(1) = 3.90, *p* = 0.048
Tastes good	82 (23.0)	14 (35.0)	96 (24.2)	*χ^2^*(1) = 2.84, *p* = 0.092
Feels good when inhaling	72 (20.2)	8 (20.0)	80 (20.2)	*χ^2^*(1) = 0.01, *p* = 0.980
Reduces my urges or craving to smoke	36 (10.1)	2 (5.0)	38 (9.6)	*χ^2^*(1) = 1.07, *p* = 0.300
Makes it easier to keep from smoking marijuana	26 (7.3)	2 (5.0)	28 (7.1)	*χ^2^*(1) = 0.29, *p* = 0.593
Keeps me from bothering other people with my smoke	38 (10.6)	10 (25.0)	48 (12.1)	*χ^2^*(1) = 6.97, *p* = 0.008
Takes away my craving to smoke faster than marijuana smoking does	22 (6.2)	2 (5.0)	24 (6.0)	*χ^2^*(1) = 0.09, *p* = 0.770
Gives me a bigger high than I can get by smoking marijuana	36 (10.1)	8 (20.0)	44 (11.1)	*χ^2^*(1) = 3.59, *p* = 0.058
Reduces withdrawal	23 (6.4)	0 (0.0)	23 (5.8)	*χ^2^*(1) = 2.74, *p* = 0.098
Helps me sleep better	58 (16.2)	17 (42.5)	75 (18.9)	***χ^2^*(1) = 16.18, *p* <0.001**
Perceived Harms of Nicotine Vape Use				
The vapor they make contains other chemicals that are not safe; they hurt my health	43 (12.0)	9 (22.5)	52 (13.1)	*χ^2^*(1) = 3.45, *p* = 0.063
I still get THC so I stay addicted	59 (16.5)	4 (10.0)	63 (15.9)	*χ^2^*(1) = 1.15, *p* = 0.284
The vapor from electronic cigarettes causes lung cancer	49 (13.7)	6 (15.0)	55 (13.9)	*χ^2^*(1) = 0.05, *p* = 0.825
The vapor from electronic cigarettes causes wet cough with mucous	48 (13.4)	4 (10.0)	52 (13.1)	*χ^2^*(1) = 0.38, *p* = 0.540
The vapor from electronic cigarettes causes dry cough	70 (19.6)	9 (22.5)	79 (19.9)	*χ^2^*(1) = 0.19, *p* = 0.664
The vapor from electronic cigarettes causes lung problems	56 (15.7)	6 (15.0)	62 (15.6)	*χ^2^*(1) = 0.01, *p* = 0.910
The vapor from electronic cigarettes causes mouth or throat cancer	59 (16.5)	2 (5.0)	61 (15.4)	*χ^2^*(1) = 3.68, *p* = 0.055
The vapor from electronic cigarettes makes me short of breath	43 (12.0)	2 (5.0)	45 (11.3)	*χ^2^*(1) = 1.78, *p* = 0.183
Using them burns my throat	64 (17.9)	10 (25.0)	74 (18.6)	*χ^2^*(1) = 1.19, *p* = 0.276
Using them gives me a dry mouth or dry throat	53 (14.8)	9 (22.5)	62 (15.6)	*χ^2^*(1) = 1.60, *p* = 0.206
Electronic cigarettes have toxic substances in them	51 (14.3)	5 (12.5)	56 (14.1)	*χ^2^*(1) = 0.10, *p* = 0.758
Electronic cigarettes don’t stop me from having urges or cravings to smoke	47 (13.2)	2 (5.0)	49 (12.3)	*χ^2^*(1) = 2.22, *p* = 0.137
I do not get enough THC/cannabis from electronic cigarettes	28 (7.8)	7 (17.5)	35 (8.8)	*χ^2^*(1) = 4.17, *p* = 0.041
It is too difficult to adjust how much THC/cannabis I get with electronic cigarettes	41 (11.5)	5 (12.5)	46 (11.6)	*χ^2^*(1) = 0.04, *p* = 0.849
Electronic cigarettes are addicting	51 (14.3)	2 (5.0)	53 (13.4)	*χ^2^*(1) = 2.68, *p* = 0.102
Electronic cigarettes do not help me quit smoking	45 (12.6)	6 (15.0)	51 (12.8)	*χ^2^*(1) = 0.18, *p* = 0.668
If I use electronic cigarettes to quit smoking, I will just go back to smoking when I stop using them	29 (8.1)	1 (2.5)	30 (7.6)	*χ^2^*(1) = 1.63, *p* = 0.202
Using electronic cigarettes gives me headaches or nausea or makes me feel dizzy	24 (6.7)	1 (2.5)	25 (6.3)	*χ^2^*(1) = 1.09, *p* = 0.297
Electronic cigarettes taste bad	16 (4.5)	8 (20.0)	24 (6.0)	***χ^2^*(1) = 15.25, *p* < 0.001**
Electronic cigarettes cause me to gain weight	14 (3.9)	1 (2.5)	15 (3.8)	*χ^2^*(1) = 0.20, *p* = 0.655

Note: With the Bonferroni correction, statistical significance was set at *p* < 0.005 for reasons for use, *p* < 0.002 for perceived benefits, and *p* < 0.003 for perceived harms. Bolded statistics indicate statistical significance.

**Table 4 ijerph-19-10392-t004:** Linear regression analyses of dependence on nicotine (PSECDI) and cannabis (PSCVDI) vaping.

	Cannabis Vape Dependence (*n* = 363)					Nicotine Vape Dependence (*n* = 403)				
	B	95% CI		β	ΔR^2^	B	95% CI		β	ΔR^2^
		LL	UL				LL	UL		
Step 1					0.07 ***					0.03 *
Age	−0.07 *	−0.13	−0.01	−0.12		−0.07 **	−0.12	−0.02	−0.14	
Bachelor’s degree vs. less	0.93	−0.26	2.13	0.09		0.79	−0.19	1.77	0.09	
Full-time employment vs. not	0.13	−1.38	1.63	0.01		−0.29	−1.65	1.06	−0.02	
Married vs. not	0.81	−0.38	2.00	0.07		0.28	−0.69	1.25	0.03	
Smoking status	1.76 **	0.54	2.98	0.15		0.33	−0.68	1.34	0.03	
Step 2					0.16 ***					0.09 ***
Years of vape use	0.13 *	0.03	0.23	0.13		0.20 ***	0.10	0.30	0.20	
Days of vape use in past 30 days	0.08 **	0.03	0.14	0.15		0.11 ***	0.07	0.15	0.27	
Place of purchase	0.26	−0.29	0.81	0.05		--	--	--	--	
State medical card vs. not	3.20 ***	2.16	4.23	0.33		--	--	--	--	
Nicotine concentration	--	--	--	--		−0.01 *	−0.01	0.00	−0.10 *	
Step 3					0.01					0.01
User type (dual vs. single)	0.11	−1.52	1.73	0.01		0.87	−0.12	1.87	0.09	

Note: CI = Confidence Interval; LL = Lower Limit; UL = Upper Limit; * *p* < 0.05, ** *p* < 0.01, *** *p* < 0.001.

**Table 5 ijerph-19-10392-t005:** Bivariate correlations assessing criterion and convergent of cannabis dependence measures.

	PSCVDI (*r*)	DSM-V (*r*)	SDS (*r*)
Criterion Validity			
DSM-V	0.66	--	--
SDS	0.49	0.56	--
VAS—How addicted are you?	0.51	0.58	0.72
Convergent Validity			
Years of cannabis vape use	0.20	0.29	0.32
Age of regular cannabis vape use	−0.23	−0.35	−0.29
Stealth vaping	0.43	0.49	0.59
THC percentage	0.30	0.32	0.32

Note: All *p*s < 0.001; PSCVDI = Penn State Cannabis Vaping Dependence Index; DSM-V = Diagnostic and Statistical Manual of Mental Disorders-V Cannabis Vape Use Disorder Score; SDS = Severity of Dependence Scale score; VAS = visual analogue scale; THC = tetrahydrocannabinol. *N*s ranged from 373–392.

## Data Availability

The data used for this manuscript are available upon reasonable request and with proper data use agreements. Please contact the corresponding author, Andrea Hobkirk, for data requests.

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
