# Peer review of "A Comparison of Vaping Behavior, Perceptions, and Dependence among Individuals Who Vape Nicotine, Cannabis, or Both"

_ijerph, 2022, doi:10.3390/ijerph191610392_

Round 1
Reviewer 1 Report
The paper addresses the question of dual nicotine and cannabis vaping by examining sociodemographic and vape use characteristics, perceived benefits and harms of use, reasons for use, and nicotine and cannabis dependence among dual and single substance vape users. The topic is highly relevant, especially considering the trends in vaping in recent years, and has not yet been examined in earlier research.
The study is well-designed and well-written. The are some minor suggestions I would propose to the authors:
Lines 604-606: Authors found that “Somewhat paradoxi- 604 cally, dual users were more likely to be combustible cigarette smokers and therefore, at 605 high risk for cigarette-related harm from toxicant exposure.” This paradoxical finding could be further elaborated on in the Discussion section. Furthermore, I suggest the Discussion section expands somewhat on the “bigger picture” of the study findings.
In addition, some minor technical issues:
Lines 168-169, 180: Add links to the references and then references number for consistency.
Lines 149-196: “… (Cuttler & Spradlin, 194 2017). The DFAQ-CU had good reliability and convergent, predictive, and discriminant 195 validity among over 2,000 cannabis users in the United States (Cuttler & Spradlin)”. Authors should be mentioned in the text, and reference numbers should be added.
Lines 258-263: “Other measures” section can be deleted from the manuscript as it might not be relevant to the reader.
Lines 288-290: Different font has been used compared to the rest of the text; please modify the font.
Lines 294-295: I suggest elaborating on why the education was recoded into two values (dichotomized).
Lines 297-298: “Marital status was categorized as married vs. not married (widowed, divorced, never 297 married, etc.)” Supplementary material (question 8) provides a category of “living with a partner”, which was put in the second category during the dichotomization, according to the authors. Considering the official statistics of unmarried couples in recent years across high-income countries, I suggest the authors either put this category together with “married” or elaborate on their original decision to the reader and leave it as it is.
Lines 440-450: I suggest adding “Cronbach’s alpha” to the lines to make it clearer to the reader.
Lines 499-500: “our findings may point to the unique demographics of vape vs. combus- 499 tible nicotine and cannabis users.”. It is unclear to the reader whether this statement refers to the unique democratic of users or the unique demographic of the Mturk sample. Please, elaborate.
Despite these minor comments, the paper is -written, clear, relevant for the field and presented in a well-structured manner. I congratulate the authors on their study and important findings.
Reviewer 2 Report
Dear Authors,
Thank you for your manuscript. The topic is important and relevant to public health. The study is complex and comprehensive. It seems that you explored everything possible on smoking. Honestly, when reading several times I had to go back as I was lost in details of procedure description, measures and findings. Possibly, I could miss some shortcomings.
It is the first paper I am reading which explains how the MTurk works. Thank you for this.
The sample is relatively small, but the study participants were selected to meet the inclusion criteria. So this is the rationale.
Below are my comments.
Abstract, line 18: what do you mean by "automated survey"?
Introduction, line 44: it would be interesting to list and discuss health risks.
Methods. Probably I missed it, but there is no information provided at which stage study participants gave their informed consent.
Also, according to the journal's formats, subsections should be numbered: 2.1., 2.2., etc. Same comment for the Results section if you decided to divide it into subsections.
